# Influence of the TAS2R38 Gene Single Nucleotide Polymorphisms in Metabolism and Anthropometry in Thyroid Dysfunction

**DOI:** 10.3390/nu15092214

**Published:** 2023-05-06

**Authors:** Marta Mendes Costa, Alda Pereira Da Silva, Carolina Santos, Joana Ferreira, Mário Rui Mascarenhas, Manuel Bicho, Ana Paula Barbosa

**Affiliations:** 1Ecogenetics and Human Health Unity, Institute for Environmental Health (ISAMB), Genetics Laboratory, Associate Laboratory TERRA, Faculty of Medicine, Lisbon University, 1649-028 Lisbon, Portugalmanuelbicho@ medicina.ulisboa.pt (M.B.);; 2Faculty of Medicine, University Clinic for General and Family Medicine, Lisbon University,1649-028 Lisbon, Portugal; 3Institute for Scientific Research Bento Rocha Cabral, Calçada Bento da Rocha Cabral 14, 1250-012 Lisbon, Portugal; 4Clinic of Endocrinology (CEDML), Diabetes and Metabolism of Lisbon Lda, 1050-017 Lisbon, Portugal; 5Faculty of Medicine, University Clinic of Endocrinology, Lisbon University, 1649-028 Lisbon, Portugal

**Keywords:** hypothyroidism, hyperthyroidism, bitter taste, TAS2R38, polymorphisms P49A/A262V/V296I

## Abstract

The gene TAS2R38 single nucleotide polymorphisms (SNPs-P49A, A262V and V296I) can condition bitter tasting by PAV (proline–alanine–valine) and non-bitter-tasting by AVI (alanine–valine–isoleucine) homozygosity. We evaluated this polymorphisms association with thyroid function, metabolism and anthropometry parameters determined by: Endpoint analysis (SNPs); DXA (fat mass-%, total fat mass—kg, lean mass—kg); Standard methods (lipid metabolism parameters, HbA1c-%, glycemia—mg/dL, insulinemia—µIU/mL, HOMA-IR, uricemia—mg/dL, calcemia—mg/dL and BMI—kg/m^2^); ELISA (leptinemia—ng/mL); Spectrophotometry (Angiotensin Converting Enzyme activity—UI/L). Statistics: SPSS program; OR [IC95%]; *p* < 0.05. Sample: 114 hypothyroid, 49 hyperthyroid, and 179 controls. An association between A262V-valine–valine and hypothyroidism/hyperthyroidism was verified (OR = 2.841; IC95% [1.726–4.676]), *p* < 0.001/OR = 8.915; IC95% [4.286–18.543]), *p* < 0.001). Protector effect from thyroid dysfunction: A262V-alanine–valine (OR = 0.467; IC95% [0.289–0.757], *p* = 0.002/OR = 0.132; IC95% [0.056–0.309], *p* < 0.001) and PAV (OR = 0.456; IC95% [0.282–0.737], *p* = 0.001/OR = 0.101; IC95% [0.041–0.250], *p* < 0.001). Higher parameter values associated with genotypes were: fat-mass-% (V296I-valine–isoleucine), lean-mass (P49A-proline–proline; PVI), leptin (AVI), HbA1c (A262V-alanine–valine) and lower values in lean-Mass (AVI; PVV), leptin (A262V-alanine–alanine), HbA1c (PVV), uricemia (V296I-valine–isoleucine), glycemia (A262V-alanine–alanine; AAV) and plasma triglycerides (PVV). In conclusion, TAS2R38 influences thyroid function, body composition and metabolism. Bitter taste perception (PAV) and the genotype A262V-alanine–valine can protect from thyroid dysfunction. AVV, PVV and genotype A262V-valine–valine may confer higher predisposition for thyroid dysfunction, particularly PVV for hyperthyroidism.

## 1. Introduction

Taste has been listed as one of the five main factors involved in food choice, along with health, cost, time, and social interactions [1]. Five different tastes have been described: sweet, bitter, umami, salted, and sour. Recently, the existence of the taste of fat has been proposed [2,3]. The chemical compounds in food activate the receptors of taste. The genotypes of these receptors, particularly their genetic single nucleotide polymorphisms (SNPs), contribute to interindividual variability in taste perception [4]. The stimuli that lead to perception of the bitter taste bind to members of the receptor protein family TAS2R, coded in the 5p, 7q, and 12p chromosomes [5].

The bitter taste receptor, type 2 member 38 (TAS2R38) is one of the members of the TAS2R family and is responsible for the perception of the bitter taste of compounds that contain the thiocyanate chemical group, such as Phenylthiocarbamide (PTC) and Propylthiouracil (PROP), which are absent in nature, but also glucosinolates present in the vegetables of the *Brassica* family [6,7]. The TAS2R38 receptor is coded by the gene TAS2R38, which is composed of 1002 nucleotides in chromosome 7 with only one exon and three single nucleotide polymorphisms: P49A, A262V and V296I [6]. These SNPs are responsible for the presence of different amino acids in the receptor protein. In this regard, P49A can lead to the presence of proline or alanine, A262V to alanine or valine, and V296I to valine or isoleucine [6]. A haplotype consists of the sequence of the three amino acids in the taste receptor protein coded by the three SNPs present in one chromosome (maternal or paternal). The association of the two haplotypes present in each chromosome comprises the TAS2R38 diplotype, which determines the expression of the receptor protein for bitter tasting (Figure 1) [8,9]. The PAV haplotype (proline–alanine–valine) is the dominant TAS2R38 variant and is associated with the bitter tasting phenotype. The recessive haplotype is the AVI variant, which is associated with the non-bitter tasting phenotype in the form of homozygosity (AVI/AVI) [6]. An individual with both PAV variants (PAV/PAV diplotype) is classified as a supertaster of the bitter taste [6]. There are also individuals with intermediate bitter tasting, which is associated with the existence of one of the haplotypes AAI, AAV or PAI with another haplotype (other than PAV), or AVI in heterozygosity with another haplotype (other than PAV) [9]. Individuals with PVV, AVV or PVI haplotypes have had no bitter tasting phenotypes described to date (Figure 1) [9].

The PAV and AVI genetic variants are described as being the most prevalent worldwide (PAV = 50.76%; AVI = 42.70%), while AAV and AAI are considered as being rare (<5% prevalence) and PAI and PVI as very rare (<1% prevalence) [10]. The PVV and AVV haplotypes only appear in a few studies in diseased samples [7,11,12,13,14].

The TAS2R38 gene is not only expressed in the gustative papillae, but is also expressed in different body locations such as the respiratory tract, genitourinary, immune and gastrointestinal systems, heart, brain, thyroid, skin, pancreas, bladder, testis and the placenta [15,16,17,18,19,20,21].

Thyroid dysfunction in general was found to be more prevalent in non-bitter tasters (68%) than in bitter tasters (32%) [22,23]. Specifically, in a sample of people with thyroid dysfunction, bitter tasting was more prevalent in those with hypothyroidism (60% of bitter tasters) than with hyperthyroidism (40% of bitter tasters) [23]. The TAS2R38 receptor is thought to influence thyroid function for the following reasons: (1) PROP is an agonist of TAS2R38 which has been indicated in the treatment of hyperthyroidism by thyroid peroxidase inhibition [24]; (2) It was found that some members of the TAS2R family (TAS2R4, TAS2R10 and TAS2R40) inhibit the TSH-dependent increase in Ca^2+^ and iodine efflux in thyrocytes [21]; (3) Non-bitter tasters demonstrated higher ingestion of bitter tasting compounds such as vegetables of the *Brassica* family, which can inhibit thyroid hormonal synthesis [7,25,26].

## 2. Materials and Methods

The study sample was randomly selected from a database of the Endocrinology, Diabetes and Metabolism Clinic of Lisbon.

Thyroid function was classified as either hypothyroidism, euthyroidism, or hyperthyroidism. The following parameters were determined and evaluated: clinical history, physical examination, serum values of fT4 (free thyroxine), fT3 (free tri-iodothyronine), TSH (thyroid-stimulating hormone), Tg Abs (thyroglobulin antibodies), TPOAbs (thyroid peroxidase antibodies) and TRAbs (TSH receptor antibodies). Individuals with hyperthyroidism consisted of 14.3% of the total sample (49 individuals), with 19 diagnosed with Graves’ disease and 30 with toxic goiter. The group with hypothyroidism comprised 33.3% of the total sample (114 individuals). All cases in the hypothyroidism sample had primary hypothyroidism, with 15 cases originated by surgical causes (8 by total thyroidectomy, 4 by subtotal thyroidectomy, and 3 by lobectomy), 21 by chronic lymphocytic thyroiditis, and 78 by non-immune causes. The control group represented 52.35% of the total sample (179 individuals) and was diagnosed with euthyroidism, having no previous clinical history of thyroid dysfunction. Diagnosis of thyroid function for all individuals in this study was made before the intake of any medication that could alter this function.

The sample of this study consisted of adult individuals with the following criteria: thyroid function classification available, oral and reading understanding of the Portuguese language, available data about the parameters in evaluation, and voluntary informed consent for study participation including DNA collection and analysis. Individuals outside of these criteria were excluded from this study.

Anthropometric parameters were determined by DXA: fat mass (%), total fat mass (kg), and lean mass (kg). The BMI (kg/m^2^) was calculated and classified according to WHO (World Health Organization) classification: underweight (<18.5 kg/m^2^); eutrophic (18.5–24.9 kg/m^2^); overweight (25–29.9 kg/m^2^); obese (≥30 kg/m^2^). Sociodemographic parameters were age and sex. Metabolic parameters were determined by standard methods of serum analysis: HbA1c (%), glycemia (mg/dL), insulinemia (µIU/mL), HOMA-IR, uricemia (mg/dL), calcemia (mg/dL) and lipid metabolism parameters (total cholesterol—mg/dl, plasma triglycerides—mg/dL, high-density lipoprotein—mg/dL and low-density lipoprotein—mg/dL). Leptinemia (ng/mL) was determined using ELISA, and Angiotensin Converting Enzyme (ACE) activity (UI/L) by using spectrophotometry.

For the genetic study, genomic DNA was first extracted in EDTA via the salting-out method, followed by the evaluation of DNA concentration and purity via spectrophotometry, and finally, the three TAS2R38 SNPs were genotyped by Endpoint genotyping. TaqMan Probes were used for each SNP. The following primers were used:

Forward Primer: CTTGGAGCAGTAAAGCAGGCTGAG

Reverse Primer: GATCTAGGCAAAGAGCTGGATGCT

The genotyping process included an initial pre-heating (95 °C, 10 min) followed by amplification of the DNA (95 °C, 5s, and then 60 °C for 1 min, 40 cycles). The SNPs were detected with a hydrolysis probe with fluorescence emission in a thermocycler.

The genetic parameters studied were the TAS2R38 haplotypes (e.g., PAV), diplotypes (e.g., PAV/PVI), and the genotype of each TAS2R38 SNPs, according to their homozygosity or heterozygosity considering both maternal and paternal chromosomes:

P49A-CG (proline–alanine) or CC (proline–proline) or GG (alanine–alanine).

A262V-CT (alanine–valine) or TT (valine–valine) or CC (alanine–alanine).

V296I-GA (valine–isoleucine) or GG (valine–valine) or AA (isoleucine–isoleucine).

The genetic additive model was used once all possible genotypes were considered, with no known dominant or recessive alleles.

Statistical analysis was performed using SPSS 25.0 for Windows (IBM, New York, NY, USA). Significance was assumed for *p*-values < 0.05. Analysis of the qualitative parameters was conducted on absolute and relative (%) frequencies with χ^2^ Pearson test. For the quantitative parameters, the Kolmogorov–Smirnov normality test was performed. Parameters with normal distributions were described as mean values ± S.D. (standard deviation) and associations were analyzed using ANOVA/Student’s *t*-test. Parameters with a non-normal distribution were described using median values (25th–75th percentiles) and their associations were analyzed with the non-parametric test of Mann–Whitney/Kruskal–Wallis. Genotypes associated with thyroid function present in the control group were evaluated by logistic multinomial regression (OR—Odds Ratio; IC 95%—confidence interval; χ^2^ Pearson test). Multiple comparison (Dunnett’s Test) was performed between thyroid function groups and between the different genotypes for each TAS2R38 SNP.

## 3. Results

The total sample comprised 342 individuals (83% women) with a mean age of 56.42 ± 13.83 years old.

### 3.1. Parameters According to Thyroid Function

#### 3.1.1. Distribution of BMI (WHO Group Classification) According to Thyroid Function

We performed a χ^2^ Pearson Test to evaluate whether there is a statistically significant difference in the categorical BMI between the groups with hypothyroidism and/or hyperthyroidism and/or the control (Table 1). A significant difference was found with a *p*-value of 0.004. Differences between each thyroid function group were studied for the BMI as a continuous variable (Table 2).

The BMI was classified according to the WHO (World Health Organization) with 1.75% of the sample being classified as underweight, 20.76% as eutrophic, 42.11% as overweight, and 3.51% as obese.

#### 3.1.2. Metabolic, Anthropometric and Sociodemographic Variables According to Thyroid Function

To understand if there was any statistically significant difference in the studied parameters between the three sample groups (hypothyroidism, hyperthyroidism, and control), we first performed an ANOVA test for age (normal distribution) and a Mann–Whitney test for the other parameters (non-normal distribution).

Thyroid function was also significantly associated with other parameters besides BMI. These associations were found for fat mass (%), total fat mass (kg), lean mass (kg), and glycemia (mg/dL) (Table 2). Significant *p*-values derived from this first analysis mean that at least one of the groups of the sample had a significantly different mean/median parameter value when comparing to the other groups of the sample.

We then proceeded to perform multiple comparison tests to determine exactly which groups (hypothyroidism, hyperthyroidism, and control) differed from each other in terms of the significant anthropometric parameters from in Table 2 (i.e., total fat mass (kg), fat mass (%) and BMI (kg/m^2^)).

Significant statistical differences were found between thyroid function groups (hypothyroidism, hyperthyroidism, and euthyroidism) for the median values of total fat mass (kg), fat mass (%), and BMI (kg/m^2^), as shown in Table 3.

### 3.2. TA2R38 Genotypes According to Thyroid Function

#### 3.2.1. TAS2R38 SNPs Genotypes According to Thyroid Function

Results obtained from the analysis of the TAS2R38 SNPs showed significant associations between thyroid function and the genotypes A626V-valine–valine (Table 4) and A626V-alanine–valine (Table 5).

#### 3.2.2. TAS2R38 Haplotypes According to Thyroid Function

The prevalence of the TAS2R38 haplotypes in the total sample was as follows: PAV (46.3%); AVI (48.1%); PVI (33%); AAV (15.3%); PVV (12.4%); AVV (3.2%); PAI (0.3%) and AAI (0.3%). The haplotype PAV was significantly associated with the absence of thyroid dysfunction (Table 6), while the presence of either PVV (Table 7) or AVV (Table 8) was associated with thyroid dysfunction.

We observed 39.82% of PAV in the group with hypothyroidism and 12.77% of PAV in the group with hyperthyroidism.

#### 3.2.3. TAS2R38 Diplotypes According to Thyroid Function

The frequency of TAS2R38 diplotypes in the total sample varied significantly with thyroid function (*p* < 0.001) by χ^2^ Pearson Test (Table 9).

### 3.3. TAS2R38 Genotypes According to Metabolic, Anthropometric and Sociodemographic (Age) Parameters

#### 3.3.1. TAS2R38 SNPs Genotypes According to Metabolic, Anthropometric and Sociodemographic (Age) Parameters

TAS2R38 SNP genotypes were significantly associated with anthropometric, metabolic and sociodemographic parameters (Table 10).

#### 3.3.2. TAS2R38 Haplotypes According to Metabolic, Anthropometric and Sociodemographic (Age) Parameters

TAS2R38 haplotypes were significantly associated with anthropometric, metabolic and sociodemographic parameters (Table 11): PAV with HbA1c; AAV with age and with glycemia; PAI with age; PVV with lean mass, plasma triglycerides, and HbA1c; AVI with lean mass and with leptin and PVI haplotype with lean mass. The only haplotype in the sample with no significant association with any of the studied parameters was the AAI haplotype.

#### 3.3.3. TAS2R38 Diplotypes According to Anthropometric and Metabolic Parameters

The only anthropometric parameters significantly associated with the TAS2R38 diplotype distribution in the total sample were BMI (*p* = 0.004) and total fat mass (*p* = 0.038). Median (25th–75th percentile) values for these parameters in the most prevalent diplotypes (Table 9) were:

PAV/AVI: total fat mass (kg) = 25.39 [20.54–30.47]; BMI (kg/m^2^) = 27.87 [25.36–31.87];

PVI/PVI: total fat mass (kg) = 25.40 [20.15–3.52]; BMI (kg/m^2^) = 28.03 [25.09–31.58];

AAV/AAV: total fat mass (kg) = 23.31 [19.53–30.10]; BMI (kg/m^2^) = 26.60 [24.03–31.94];

PVV/AVI: total fat mass (kg) = 24.54 [21.28–28.39]; BMI (kg/m^2^) =27.10 [25.30–30.90].

Higher values for both total fat mass and BMI were obtained for the PAV/PVI diplotype when compared with the other diplotypes in total sample (PAV/PVI: total fat mass (kg) = 30.05 [26.60–35.71]; BMI (kg/m^2^) = 31.62 [29.15–34.99]).

## 4. Discussion

The anthropometric and body composition parameters (Table 2) obtained reflected the decreased basal energetic expenditure described for hypothyroidism, and the opposite phenomenon for hyperthyroidism.

Regarding the analysis of the TAS2R38 SNPs in thyroid function, the SNP A262V was the only SNP with significant influence, with the genotype A262V-valine–valine associated with the presence of thyroid dysfunction (hypo- or hyperthyroidism) and in opposition, the genotype A262V-alanine–valine associated with the absence of thyroid dysfunction (Table 4 and Table 5).

It is important to mention that, as the controls were recruited in a metabolic clinic, it raises the suspicion that they were not entirely free of disease. Despite this, we are certain that these controls did not suffer from thyroid dysfunction, which was the object of this study. In fact, the controls may suffer from other diseases that may or not be correlated with the prevalence of PAV in this sample, which could be addressed in future studies.

The fact that the frequency of some usually rare haplotypes was non-negligible in the samples with thyroid dysfunction is in line with previously published studies in diseased samples [7,11,12,13,14], and not in the global population.

Additionally, the PVV and AVV haplotypes (both with unknown bitter tasting phenotypes) [9] showed a statistically significant difference between the sample groups (hypothyroidism and/or hyperthyroidism and/or control), both with higher prevalence in hyperthyroidism and absence in the control group (Table 7 and Table 8).

On the other hand, PAV (bitter tasting haplotype) [6] showed a significantly lower risk of thyroid dysfunction according to the multiple regression (Table 6).

We suggest that the statistically significant prevalence of this haplotype could derive from the contribution of bitter food ingestion (such as vegetables of the *Brassica* family) [17,25,26] and/or cellular expression of TAS2R38 in thyrocytes [21] to thyroid function. This suggestion warrants further study.

Considering the possible contribution of both food ingestion and TAS2R38 cellular expression (in thyrocytes) to thyroid function, the higher tendency of PVV and AVV for either hypo- or hyperthyroidism could vary according to the degree of influence of each one of these two factors in thyroid function. In this way, a higher tendency to develop hypothyroidism could be due to increased ingestion of bitter goitrogenic compounds with an ability to inhibit thyroid function [25,26,27,28]. In this way, the ingestion of these bitter compounds in individuals with the PVV and/or AVV haplotypes could indicate a lower perception of bitter taste in these haplotypes. However, this hypothesis needs to be investigated in further studies, as there is not yet a description of the bitter tasting phenotypes associated with these haplotypes [9].

In relation to hyperthyroidism, its predominance associated with PVV and/or AVV (Table 7 and Table 8) could be mainly due to the possibility of TAS2R38 expression in thyrocytes with lower sensitivity to antithyroid effects exerted by the exposition of individuals with these haplotypes to TAS2R38 agonists [21]. This high thyroid hormonal production could be explained by three possible mechanisms: less thyroid peroxidase (TPO) inhibition; higher deiodinases activity and decreased inhibition of the TSH-dependent Ca^2+^ signaling and/or iodine efflux in the thyrocytes [21]. The first two mechanisms mentioned could be related to hyperthyroidism propension because TAS2R agonists such as propylthiouracil, approved for hyperthyroidism treatment [24,29], can inhibit TPO [21] and type 1 deiodinase activity [30,31,32]. As for the increased TSH-dependent Ca^2+^ and iodine efflux in thyrocytes, it has been shown that some members of the TAS2R family (TAS2R4, TAS2R10 and TAS2R40) inhibit both of these mechanisms [21].

The different TAS2R38 SNPs and haplotypes influenced anthropometry and metabolism (Table 10 and Table 11). An increase in lean mass median value was verified for both PVI haplotype and P49A-proline–proline genotype. In contrast, the AVI haplotype showed a lower lean mass median value but also a higher median value for leptin, which could be related to behaviors of higher food disinhibition [33] and higher fat-rich food consumption [34,35,36,37,38], both described for this haplotype.

On the other hand, leptin showed a significantly lower median value for the A262V-alanine–alanine genotype; this is possibly associated with higher levels of satiety in individuals with this genotype in line with the lower values of leptin associated with the “T” allele (valine) of the SNP A26V in a previous study [33]. When compared to other haplotypes in the total sample, the AAV haplotype (intermediate bitter taster) showed a significantly lower glycemic median value. Individuals with this haplotype also had a lower mean age value compared to other TAS2R38 haplotypes. Lower glycemic values can lead to a lower risk of diabetes, which has been previously associated with the bitter tasting phenotype [39,40]. On the contrary, the prevalence of younger individuals with this haplotype can indicate a propensity towards less longevity, which has been previously associated with the non-bitter tasting phenotype [19]. In this way, considering that the AAV haplotype corresponds to an intermediate bitter tasting phenotype, it could present results associated with both bitter and non-bitter tasting phenotypes.

Fat mass (%) and uricemia were associated with only one SNP genotype, namely the V296I-valine–isoleucine. This genotype, despite showing higher median value for fat mass (%), also showed an inferior median value for uricemia when compared to other SNP V26I genotypes. This result for uricemia is thought to be not correlated directly with fat body mass but to other risk factors for hyperuricemia that can possibly be diminished for this genotype, such as ingestion of red meat, sweet beverages, seafood, and fructose, among other factors [41,42,43,44].

The PVV haplotype showed significant associations with parameters that can be related to its higher prevalence in the group diagnosed with hyperthyroidism. Such results were lower median values for lean mass, Hb1Ac, and plasma triglycerides. In hyperthyroidism, there is a known increase in basal energetic expenditure, which could be related to a reduction in lean mass [45,46,47,48]. The lower Hb1Ac (without significant difference in glycemic values) and plasma triglyceride median values observed in the PVV haplotype could be caused by an increase in erythropoiesis [49,50] and higher lipoprotein lipase activity with higher removal rate of triglycerides from blood circulation [51,52,53], respectively, which could result from the prevalence of PVV in hyperthyroidism.

Of the eleven TAS2R38 diplotypes observed in the total sample, four corresponded to bitter tasting phenotypes (PAV/PVI; PAV/AVI; PAV/AAV; PAV/AAI) and one corresponded to an intermediate bitter tasting phenotype (AAV/AAV). The remaining TAS2R38 diplotypes were composed of PVI and/or PVV and/or AVV and a haplotype other than PAV, resulting in unknown bitter tasting phenotypes. There were no non-bitter taster phenotypes in the sample, as no AVI/AVI homozygous genotypes were detected. Regarding the prevalence of bitter tasting (PAV) in the thyroid dysfunction sample, the results from this study were in line with the ones described in a previous study [23], since most bitter tasters in the thyroid dysfunction sample were observed in hypothyroidism when compared to hyperthyroidism.

TAS2R38 diplotypes were associated with thyroid function (Table 9). The PVV/AVI and AVV/AVV diplotypes were absent from the control group, with the PVV/AVI showing a higher prevalence in thyroid dysfunction, mainly in hyperthyroidism. The most prevalent TAS2R38 diplotype in the control group was PAV/AVI, which was associated with euthyroidism. The influence of these diplotypes on thyroid function can be related to the changes in the anthropometric parameters [54,55,56,57,58], with PVV/AVI having lower BMI, total fat mass, and lean mass median values when compared to the PAV/AVI diplotype. These results can be related to the higher metabolic rate that occurs in hyperthyroidism [59,60,61], since the PVV/AVI diplotype was most prevalent in the sample with hyperthyroidism than in the sample with hypothyroidism. Additionally, the PAV/PVI diplotype showed higher total fat mass, and BMI median values comparing to the other TAS2R38 diplotypes in the sample, which could be related to its absence from the hyperthyroidism sample (Table 9).

A limitation of this study is that we only analyzed the association between the studied parameters and thyroid function. In future studies, it would be interesting to discriminate these associations according to the underlying pathology that leads to thyroid dysfunction (these data were not provided for this study).

In future studies, it would be of value to analyze the food consumption patterns associated with different TAS2R38 genotypes and corresponding bitter tasting phenotypes. It would also be of interest to study the contribution of hormonal thyroid synthesis to the origin and progression of thyroid dysfunction in the TAS2R38 genotypes. Finally, aligned with this purpose, it would be useful to obtain more knowledge about the variety of compounds with agonist action in the TAS2R38 receptors expressed in the thyroid.

## 5. Conclusions

The TAS2R38 genotype showed an association with thyroid function, body composition and metabolism. The only TAS2R38 SNP with significant association with thyroid function was the A262V.

The results from this study indicate that bitter taste perception (PAV) and the genotype (A262V-alanine–valine) present a lower risk of thyroid dysfunction. Instead, the TAS2R38 haplotypes AVV and PVV had statistically significant different distribution in the samples studied, showing absence from the controls.

The SNP A262V-valine–valine genotype presented a higher risk for thyroid dysfunction according to the multiple regression analysis performed.

However, the absence of homozygous non-bitter tasters (AVI/AVI) and the presence of unknown bitter perception for PVV and AVV haplotypes limits the comprehension of this genotype’s influence on the studied parameters. In conclusion, genetic variation of TAS2R38 showed a significant association with thyroid function with modulation of satiety and metabolism associated with body composition and anthropometry.

## Figures and Tables

**Figure 1 nutrients-15-02214-f001:**
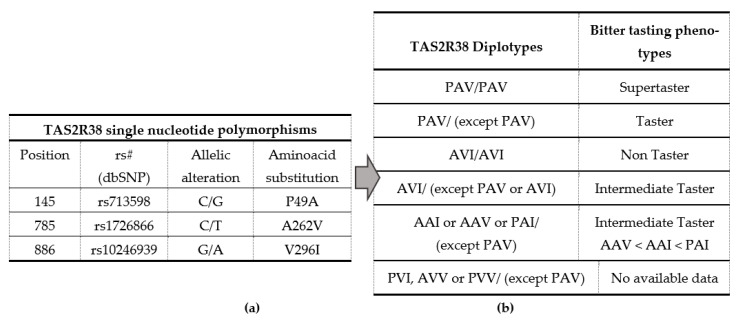
Tasting genotype–phenotype correspondence. (**a**) TAS2R38 SNPs genotypes; *dbSNP*: The Single Nucleotide Polymorphism Database; rs#: reference SNP ID number; P = proline; A = alanine; V = valine; I = isoleucine; (**b**) TAS2R38 diplotypes and correspondent bitter tasting phenotypes [6,9].

**Table 1 nutrients-15-02214-t001:** Distribution of BMI (WHO group classification) according to thyroid function.

BMI (kg/m^2^) WHO Classification	Hypothyroidism N (%)	Hyperthyroidism N (%)	Control N (%)	Total Sample N (%)
<18.5	0 (0.00)	2 (4.08)	4 (2.23)	6 (1.75)
18.5–24.9	19 (16.67)	17 (34.69)	35 (19.55)	71 (20.76)
25–29.9	49 (42.98)	24 (48.98)	71 (39.66)	144 (42.11)
≥30	46 (40.35)	6 (12.24)	69 (38.55)	121 (35.38)
Total	114 (100)	49 (100)	179 (100)	342 (100)

N: number of individuals in each BMI group; %: Percentage of individuals in each BMI group; WHO (World Health Organization); BMI: body mass index; *p*-χ^2^ Pearson Test; Significance: *p* < 0.05.

**Table 2 nutrients-15-02214-t002:** Statistically significant associations between the presence or absence of hypothyroidism/hyperthyroidism and metabolic, anthropometric and sociodemographic (age) parameters.

Parameters	Thyroid Function	N	Mean ± S.D./Median [Percentile 25th–75th]	*p*-Value
Total fat mass (kg)	Hypothyroidism	109	26.21 [21.63–30.99]	0.004
	Hyperthyroidism	48	22.76 [17.16–7.68]	
	Control	171	24.98 [19.94–30.93]	
	Total	328	25.08 [20.31–30.45]	
Lean mass (kg)	Hypothyroidism	109	42.80 [39.37–47.47]	0.019
	Hyperthyroidism	48	39.99 [17.16–5.92]	
	Control	171	43.25 [38.73–50.87]	
	Total	328	42.80 [38.65–48.96]	
Fat mass (%)	Hypothyroidism	109	38% [35–42%]	0.025
	Hyperthyroidism	48	35% [30–40%]	
	Control	171	37% [31–41%]	
	Total	328	38% [32–41%]	
BMI (kg/m^2^)	Hypothyroidism	109	28.76 [26.03–31.93]	<0.001
	Hyperthyroidism	48	25.85 [22.42–25.85]	
	Control	171	28.14 [25.20–32.19]	
	Total	328	27.89 [25.19–31.58]	
Age (years)	Hypothyroidism	114	58.46 ± 12.50	0.036
	Hyperthyroidism	49	56.21 ± 13.81	
	Control	179	56.21 ± 13.81	
	Total	342	56.42 ± 13.83	
Glycemia (mg/dL)	Hypothyroidism	98	84.00 [76.75–93.00]	0.014
	Hyperthyroidism	33	82.00 [79.00–92.00]	
	Control	157	86.00 [77.00–103.00]	
	Total	288	84.00 [77.00–97.00]	

N: Valid number of individuals for each analyzed parameter; BMI: body mass index (kg/m^2^); *p*-ANOVA (age) and p-test of Mann–Whitney (all other parameters); S.D.: Standard Deviation; Significant values for *p* < 0.05. Only parameters with significant association with thyroid function are shown.

**Table 3 nutrients-15-02214-t003:** Statistically significant differences between hypothyroidism, hyperthyroidism and control for anthropometric, metabolic and sociodemographic (age) parameters.

Parameters	Thyroid Function	Median Difference	S.D.	*p*-Value
Total fat mass (kg)	Hypothyroidism–Hyperthyroidism	+0.93	1.14	0.001
	Hyperthyroidism–Control	−3.07	1.10	0.019
Fat mass (%)	Hypothyroidism–Hyperthyroidism	+2.94	1.11	0.029
BMI (kg/m^2^)	Hypothyroidism–Hyperthyroidism	+3.51	0.72	<0.001
	Hyperthyroidism–Control	−3.07	0.70	<0.001
Lean mass (kg)	Hyperthyroidism–Control	−3.83	1.20	0.005

S.D.: Standard deviation; BMI: Body mass index (kg/m^2^); *p*-Dunnett’s Test; Statistical significance for *p* < 0.05.

**Table 4 nutrients-15-02214-t004:** Distribution of the genotype SNP A262V-valine–valine of the TAS2R38 gene according to thyroid function.

Sample According to Thyroid Function	A262V Alanine–Valine or Alanine–Alanine N (%)	A262V Valine–Valine N (%)	OR; [IC95%]; (*p*-Value)
Hypothyroidism	57 (50.4)	56 (49.6)	2.841; [1.726–4.676]; (<0.001)
Hyperthyroidism	12 (24.5)	37 (75.5)	8.915; [4.286–18.543]; (<0.001)
Control	133 (74.3)	46 (25.7)	Reference
Total	202 (59.2)	139 (40.8)	___

N: Valid number of individuals for the analyzed parameter; %: percentage of the TAS2R38 genotype in hypo/hyperthyroidism, control, or total sample. OR: Odds Ratio by logistic multinomial regression with statistical significance for *p* < 0.05; IC: confidence interval.

**Table 5 nutrients-15-02214-t005:** Distribution of the genotype SNP A262V-alanine–valine of the TAS2R38 gene according to thyroid function.

Sample According to Thyroid Functions	A262V Alanine–Alanine or Valine–Valine N (%)	A262V Alanine–Valine N (%)	OR; [IC95%]; (*p*-Value)
Hypothyroidism	71 (62.8)	42 (37.2)	0.467; [0.289–0.757] (0.002)
Hyperthyroidism	42 (85.7)	7 (14.3)	0.132; [0.056–0.309] (0.001)
Control	79 (44.1)	100 (55.9)	Reference
Total	192 (56.3)	149 (43.7)	____

N: Valid number of individuals for the analyzed parameter; %: percentage of the TAS2R38 genotype in hypo/hyperthyroidism, control, or total sample. OR: Odds Ratio by logistic multinomial regression with statistical significance for *p* < 0.05; IC: confidence interval.

**Table 6 nutrients-15-02214-t006:** Distribution of the PAV (proline–alanine–valine) haplotype in the sample according to thyroid function.

Sample According to Thyroid Function	Non-PAV Haplotypes N (%)	PAV N (%)	OR; [IC95%]; (*p*-Value)
Hypothyroidism	68 (60.21)	45 (39.8)	0.456; [0.282–0.737]; (0.001)
Hyperthyroidism	41 (87.2)	6 (12.8)	0.101; [0.041–0.250]; (0.001)
Control	73 (40.8)	106 (59.2)	Reference
Total	182 (53.7)	157 (46.3)	____

N: Valid number of individuals for the analyzed parameter; %: percentage of the TAS2R38 haplotype in hypothyroidism, hyperthyroidism, control, or total sample. Statistical significance for *p* < 0.05; OR: Odds Ratio by logistic multinomial regression; IC: confidence interval.

**Table 7 nutrients-15-02214-t007:** Distribution of the PVV (proline-valine–valine) haplotype in the sample according to thyroid function.

Sample According to Thyroid Function	Non-PVV Haplotypes N (%)	PVV N (%)	*p*-Value
Hypothyroidism	94 (83.2)	19 (16.8)	0.001
Hyperthyroidism	24 (51.1)	23 (48.9)	
Control	179 (100)	0 (0.0)	
Total	297 (87.6)	42 (12.4)	

N: Number of valid individuals for the analyzed parameter; %: percentage of the TAS2R38 haplotype in hypothyroidism, hyperthyroidism, control or total sample; *p*-χ^2^ Pearson Test; Statistical significance for *p* < 0.05.

**Table 8 nutrients-15-02214-t008:** Distribution of the AVV (alanine–valine–valine) haplotype in the sample according to thyroid function.

Sample According to Thyroid Function	Non-AVV Haplotypes N (%)	AVV N (%)	*p*-Value
Hypothyroidism	107 (94.7)	6 (5.3)	0.001
Hyperthyroidism	42 (89.4)	5 (10.6)	
Control	179 (100)	0 (0.0)	
Total	328 (96.8)	11 (3.2)	

N: Number of valid individuals for the analyzed parameter; %: percentage of the TAS2R38 haplotype in hypothyroidism, hyperthyroidism, control or total sample; *p*-χ^2^ Pearson Test; Statistical significance for *p* < 0.05.

**Table 9 nutrients-15-02214-t009:** TAS2R38 distribution according to thyroid function.

Sample According to Thyroid Function	PVI/ PVI N (%)	PVV/ PVI N (%)	PVV/ AVI N (%)	PVV/ AVV N (%)	AVV/ AVV N (%)	PAI/ PVI N (%)	PAV/ PVI N (%)	PAV/ AVI N (%)	AAV/ AAV N (%)	PAV/ AAV N (%)	PAV/ AAI N (%)
Hypothyroidism	32	0	18	1	5	0	8	34	12	3	0
	(28)	(0)	(15.9)	(0.9)	(4.4)	(0)	(7.1)	(30.1)	(10.6)	(2.7)	(0)
Hyperthyroidism	10	3	18	2	3	0	0	6	5	0	0
	(21.3)	(6.4)	(38.3)	(4.3)	(6.4)	(0)	(0)	(12.8)	(10.6)	(0)	(0)
Control	46	0	0	0	0	1	12	87	26	6	1
	(25.7)	(0)	(0)	(0)	(0)	(0.6)	(6.7)	(48.6)	(14.5)	(3.4)	(0.6)
Total	88	3	36	3	8	1	20	127	43	9	1
	(26)	(0.9)	(10.6)	(0.9)	(2.4)	(0.3)	(5.9)	(37.5)	(12.7)	(2.7)	(0.3)

N: Number of valid individuals for the analyzed parameter; %: TAS2R38 diplotype percentage for each thyroid function group, and total sample; Statistical significance for *p* < 0.05.

**Table 10 nutrients-15-02214-t010:** Statistically significant associations between TAS2R38 SNPs and metabolic, anthropometric, and sociodemographic (age) parameters.

TAS2R38 SNPs	Parameters	SNPs Genotypes	Mean ± S.D./Median [Percentile 25th–75th]	*p*-Value
P49A	Lean mass (kg)	P49A-proline–proline	43.53 [39.86–51.33]	0.031
		Non-P49A-proline–proline	42.48 [38.24–47.43]	
A626V	Leptin (ng/dL)	A626V-alanine–alanine	23.59 [14.10–52.62]	0.037
		Non-A626V-alanine–alanine	37.05 [22.74–66.78]	
	Age (years)	A626V-alanine–alanine	52.31 ± 13.19	0.029
		Non-A626V-alanine–alanine	57.12 ± 13.87	
	Glycemia (mg/dL)	A626V-alanine–alanine	78.00 [76.00–89.50]	0.020
		Non-A626V-alanine–alanine	85.00 [79.00–98.00]	
	HbA1c (%)	A626V-alanine–valine	4.60 [3.60–6.33]	0.030
		Non-A626V-alanine–valine	4.00 [3.55–5.20]	
V296I	Age (years)	V296I-valine–valine other	53.18 ± 13.29	0.046
		Non-V296I-valine–valine other	57.02 ± 13.84	
	Uricemia (mg/dL)	V296I-valine–isoleucine	4.58 ± 1.33	0.026
		Non-V296I-valine–isoleucine	5.03 ± 1.42	
	Fat mass (%)	V296I-valine–isoleucine	38% [32–41%]	0.048
		Non-V296I-valine–isoleucine	37% [31–41%]	

S.D.: Standard Deviation; *p*-Student’s *t*-test (age) or Kruskal–Wallis test; Significant values for *p* < 0.05.

**Table 11 nutrients-15-02214-t011:** Statistically significant associations found between TAS2R38 haplotypes and metabolic, anthropometric and sociodemographic (age) parameters.

Parameters	TAS2R38 Haplotypes	Mean ± S.D./Median [Percentile 25th–75th]	*p*-Value
HbA1c (%)	PAV	5.60 [3.60–7.30]	0.048
	Non-PAV	4.00 [3.60–5.20]	
	PVV	3.10 [3.05–3.90]	0.025
	Non-PVV	4.90 [3.60–6.45]	
Plasma triglycerides (mg/dL)	PVV	80.00 [66.00–98.00]	0.038
	Non-PVV	98.00 [74.00–126.00]	
Glycemia (mg/dL)	AAV	78.50 [76.00–89.75]	0.027
	Non-AAV	85.00 [79.00–98.00]	
Age (years)	AAV	52.98 ± 13.31	0.033
	Non-AAV	56.98 ± 13.8	
	PAI	24.60	0.021
	Non-PAI	56.4 ± 13.72	
Leptin (ng/mL)	AVI	42.25 [26.31–65.41]	0.019
	Non-AVI	29.79 [16.32–62.48]	
Lean mass (kg)	PVV	41.00 [36.86–45.31]	0.036
	Non-PVV	43.01 [38.72–49.38]	
	AVI	42.12 [38.50–46.75]	0.043
	Non-AVI	43.25 [38.89–50.97]	
	PVI	43.48 [39.83–51.35]	0.039
	Non-PVI	42.49 [38.18–47.46]	

S.D.: Standard Deviation; *p*-Student’s *t*-test (age) or Kruskal–Wallis test; Significant values for *p* < 0.05.

## Data Availability

The datasets used and or analyzed during the current study are available from the corresponding author upon reasonable request.

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
