# Peer review of "Influence of the TAS2R38 Gene Single Nucleotide Polymorphisms in Metabolism and Anthropometry in Thyroid Dysfunction"

_nutrients, 2023, doi:10.3390/nu15092214_

Round 1
Reviewer 1 Report
The manuscript by Mendes Costa and collaborators sought to investigate the association between TAS2R38 polymorphisms and thyroid function in a cohort from a metabolic clinic in Lisbon, Portugal. There is not much literature on the relationship between taste receptor variants and thyroid disease, what makes this study potentially interesting. However, the presentation of methods and data, in my opinion, are quite poor and the whole text needs a thorough editing.
Major criticism
Page 3. In the methods there is some confusion about the classification of thyroid diseases. Apart from choosing a functional criterion (hyper-, hypothyroidism) rather than the underlying pathology (autoimmune, non-autoimmune thyroiditis, cancer, etc.) some categories are ambiguous. For instance, what was the cause of thyroidectomy? Toxic goiter, cancer? Were the euthyroids former hypothyroid patients who later took levo-thyroxine? This is a significant limitation that should be corrected and, if not possible, acknowledged in the discussion.
Page 4. The table 1 contains several errors. First, in the sixth column of the total, the number corresponding to the highest BMI category is not 12 but 121, therefore the corresponding percentage is 35.38 (it is also not clear why two decimals were used, whereas integers were used in the other columns). More importantly, only the statistical comparison between the hyperthyroidism column with controls gives a significant value (p=0.004), while the comparison of the hypothyroidism column with controls (0,19,49,46 vs 4,35,71 ,69) is not significant at all (p=0.375).
Page 5. In table 2, the p-value likely derives from the multiple comparison test, but it is unclear which subgroup comparisons it refers to. For example, the average BMI doesn't seem much different between the hypothyroid subgroup and controls, so what does the p-value of <0.001 refer to? Same for lean mass, etc.
Page 6. In Table 4 there are mistakes in the calculation of the confidence intervals of OR, e.g. in the case of hypothyroidism the upper limit is 4.676 not 0.676, in the case of hyperthyroidism the upper limit is 18.543 not 8.543.
Page 9. In my opinion, the first twelve lines of the discussion express rather trivial considerations and may be shortened.
Page 10. The discussion contains questionable considerations. The fact that the frequency of the PAV haplotype was more frequent among controls obviously raises the question of how controls were selected. The fact that they were recruited in a metabolic clinic may lead to the suspicion that they were not entirely free from disease, which was not sufficiently detailed in the materials and methods section. The fact that the frequency of some usually rare haplotypes was non-negligible raises the issue of the quality of genotyping, which has not been confirmed by DNA sequencing. The considerations in lines 17-34 are purely speculative. Finally, the authors in the conclusions cannot use expressions (influence, protect) that insinuate a causal inference, which is not legitimate given the study design.
Minor remarks
The text is quite readable, but it was difficult to refer to specific sentences due to the absence of line numbering.
Page 3, materials and methods, line 19. The acronym ADN should be written as DNA.
Page 3, lines 23-24. If possible, each BMI category should be accompanied by the unit of measurement, kg/m². In line 26, the unit of measure for insulin levels should be µIU/mL.
Page 3, line 29. What is ACE? Angiotensin Converting Enzyme ?
Page 4, line 1. Some more details about the PCR reaction would be welcome (oligoprimer). Moreover, in line 4 the “thermocirculator” must be called “thermocycler”.
Page 11. “triglyceridemia” or better, “plasma triglycerides”
Author Response
Dear Reviewer,
Please see the attachment. This second attachment is the article with the corrections.
Thank you
Best Regards

Reviewer 2 Report
This study explored the association between SNPs in the TAS2R38 gene and thyroid function, metabolism, and body composition. The TAS2R38 gene is responsible for the perception of bitter taste and has three SNPs that determine the expression of the receptor protein for bitter tasting.
The study found that individuals with the A262V-valine-valine genotype were more likely to have hypothyroidism/hyperthyroidism, while A262V-alanine-valine and PAV genotypes were protective against thyroid dysfunction. The study also found that different genotypes were associated with higher or lower values for various parameters, including fat mass, lean mass, HbA1c, and triglyceride levels.
Overall, the study suggests that TAS2R38 influences thyroid function, body composition, and metabolism, and that bitter taste perception and certain genotypes may confer protection or higher predisposition for thyroid dysfunction.
Table 2. Suggest to notify in the legend that p-value is for comparison between the top 3 and not included the total values. Looks like multiple comparison was not performed for this table.
My major concern: as SNP papers, the outlay of data presentation is not familiar, which hinders analyzing the data as the five genetic association models.
Author Response
Dear reviewer,
Please see the attachment.
Best Regards
Thank you,
Best Regards

Round 2
Reviewer 1 Report
Taking into account my remarks the authors have answered satisfactorily and made substantial changes to the manuscript which now appears more accurate. I have nothing else to ask.